# A Narrative Review of the Usefulness of Indocyanine Green Fluorescence Angiography for Perfusion Assessment in Colorectal Surgery

**DOI:** 10.3390/cancers14225623

**Published:** 2022-11-16

**Authors:** Masayoshi Iwamoto, Kazuki Ueda, Junichiro Kawamura

**Affiliations:** Department of Surgery, Faculty of Medicine, Kindai University, 377-2, Ohnohigashi, Osaka Sayama 589-8511, Japan

**Keywords:** colorectal surgery, indocyanine green, near-infrared, fluorescence imaging, fluorescence angiography, perfusion, anastomotic leakage

## Abstract

**Simple Summary:**

Anastomotic leakage is one of the most dreaded complications of colorectal surgery and adequate perfusion at the anastomotic site is a well-recognized prerequisite to prevent it. Indocyanine green fluorescence angiography (ICG-FA) is a novel technology that allows real-time assessment of tissue perfusion and has become widely used in practice as it is feasible to use. However, there is no consistent evidence as to whether this new technology reduces anastomotic leakage after colorectal surgery and the results of large-scale randomized controlled trials currently underway are awaited. In addition, several methods have been proposed to objectively evaluate fluorescence using various quantitative parameters. In this review, we focus on the utility of ICG-FA in reducing postoperative anastomotic leakage in colorectal surgery, with the intention of providing an up-to-date information and discussing future perspectives in this field.

**Abstract:**

Anastomotic leakage is one of the most dreaded complications of colorectal surgery and is strongly associated with tissue perfusion. Indocyanine green fluorescence angiography (ICG-FA) using indocyanine green and near-infrared systems is an innovative technique that allows the visualization of anastomotic perfusion. Based on this information on tissue perfusion status, surgeons will be able to clearly identify colorectal segments with good blood flow for safer colorectal anastomosis. The results of several clinical trials indicate that ICG-FA may reduce the risk of AL in colorectal resection; however, the level of evidence is not high, as several other studies have failed to demonstrate a reduction in the risk of AL. Several large-scale RCTs are currently underway, and their results will determine whether ICG-FA is, indeed, useful. The major limitation of the current ICG-FA evaluation method, however, is that it is subjective and based on visual assessment by the surgeon. To complement this, the utility of objective evaluation methods for fluorescence using quantitative parameters is being investigated. Promising results have been reported from several clinical trials, but all trials are preliminary owing to their small sample size and lack of standardized protocols for quantitative evaluation. Therefore, appropriately standardized, high-quality, large-scale studies are warranted.

## 1. Introduction

Colorectal cancer is the third most common malignancy and the second leading cause of cancer-related mortality worldwide [1]. Although advances in surgical techniques and the use of neoadjuvant and adjuvant therapies have improved the long-term prognosis of colorectal cancer, surgery remains the cornerstone of treatment for this condition.

Complete mesocolic excision (CME) with central vascular ligation (CVL), introduced by Hohenberger in 2009, and total mesocolic excision (TME), introduced by Heald in 1982, are regarded as standard concepts for colon and rectal cancer surgery, respectively [2,3,4,5,6]. The use of minimally invasive surgery has become widespread over the past two decades because the magnified field of view and meticulous dissection in laparoscopic and robotic surgery are useful for achieving sharp dissection of the embryological plane common to CME and TME, and better short-term outcomes and reasonable long-term oncological results have been confirmed [7,8,9,10,11]. Furthermore, using minimally invasive surgery for rectal cancer, a significant proportion of abdominoperineal resections appear to have been replaced by sphincter-preserving surgery [12,13].

However, minimally invasive surgery is associated with the lack of tactile sensation that facilitates the identification of tumors and anatomical structures. Intraoperative visualization techniques, such as near-infrared (NIR) fluorescence imaging, are attracting interest as new technologies to compensate for this. Fluorescence imaging requires fluorescent dyes which highlight specific tissues and fluorescence-capable camera systems. Fluorescent dyes used can be classified as either specific or non-specific. Specific dyes are fluorescently labeled antibodies or nanoparticles that have high affinity for the target of interest and have the advantage of high selectivity of target tissues such as tumors [14]. However, they are more expensive than non-specific dyes, tend to emit weaker fluorescence, and require approval from national regulatory boards before they can be used in clinical practice because they are novel agents. On the other hand, non-specific dyes are not selective to specific tissues, but can highlight tissues with fluorescence when perfused into tissues or excreted from the liver or kidney. Indocyanine green (ICG) and methylene blue, representatives of non-specific dyes, have been used in clinical practice for many years with excellent safety profiles, and are commonly used for fluorescence imaging. In particular, ICG fluorescence navigation surgery has become widely used in clinical practice because it is safe, easy to use, real-time, and highly cost-effective, in addition to having the best signal-to-noise ratio.

ICG has been approved by the U.S. Food and Drug Administration (FDA) for clinical and research use in humans since 1956. The compound is water-soluble, and when administered into the blood, it binds rapidly and extensively to plasma proteins, trapping them in intravascular compartments with minimal leakage into the interstitium. In the 1970s, it was discovered that protein-bound ICG emits fluorescence peaking at approximately 830 nm when irradiated with NIR light of approximately 750–800 nm wavelength [15]. As the human eye cannot detect light at this wavelength, the fluorescent agent does not interfere with the standard surgical field and can be used for intraoperative imaging with appropriate equipment. Because ICG solutions contain trace amounts of iodine, they are absolutely contraindicated in cases of proven allergy to iodides such as contrast agents for computed tomography. However, the incidence of serious adverse reactions, such as anaphylactic reactions due to ICG administration, has been reported to be <0.05%, indicating a high level of safety [16].

ICG-based real-time imaging was first clinically applied in the 1990s in the field of ophthalmology to evaluate retinal vessels [17]. Since then, this technology has been widely used for navigation in surgery [18,19]. In the field of colorectal surgery, it has been applied in the assessment of tissue perfusion and vasculature, lymphatic drainage [20,21,22], assessment of tumors [23,24,25], and identification of the urinary tract [26,27]. In this review, we discuss the clinical utility of ICG-based intra-operative imaging, focusing on the assessment of tissue perfusion at the anastomotic site, which is now widely used in clinical practice.

## 2. Assessment of Tissue Perfusion at the Anastomotic Site

### 2.1. Anastomotic Leak in Colorectal Surgery

Anastomotic leakage (AL) is one of the most serious complications of colorectal resection. The incidence of AL is 1–19%, and its frequency increases in distal anastomoses such as colorectal and coloanal anastomoses [28,29,30]. AL increases the perioperative morbidity and mortality [31], length of hospital stay, and medical cost [32,33]. It often results in reoperation and the need for a temporary or definitive stoma, and consequently, has a significant impact on the patient’s quality of life [28]. Moreover, AL is associated with decreased disease-free and overall survival as it significantly increases the rate of local tumor recurrence in patients with cancer [34,35,36].

In general, AL is defined as any defect in the intestinal wall at the anastomosis leading to a communication between the intra- and extra-luminal components. As proposed by the International Study Group of Rectal Cancer, the severity of AL is commonly classified into three grades: grade A did not require any intervention; grade B required active intervention without the need for reoperation; and grade C required reoperation [37].

Several risk factors for AL have been reported. Patient factors include age, male sex, smoking, diabetes mellitus, American Society of Anesthesiologists (ASA) score 2 or higher, obesity, poor nutrition, alcohol abuse, use of steroids or anticoagulants, intraoperative septic conditions, preoperative chemotherapy, radiation therapy, tumor size, tumor location, and malignant disease; surgical factors include procedural proficiency, blood loss and transfusion, tension on the anastomosed intestine, and anastomotic tissue perfusion [28,29,30]. As the development of AL is multifactorial, it is difficult to accurately predict the perioperative risk of AL; however, surgical factors are modifiable in some cases. Among them, adequate perfusion is a well-recognized prerequisite for complete healing of intestinal anastomoses.

### 2.2. Perfusion Assessment in Colonic Stumps

Historically, colorectal surgeons have assessed blood perfusion of the intestinal stump and anastomosis site by the color of the intestinal wall, visible peristalsis, bleeding from the marginal artery after intestinal transection, and tactile detection of mesenteric artery pulsation. However, the clinical assessment of anastomotic blood perfusion using these classical methods is highly subjective and lacks sufficient predictive accuracy [38]. Recently, various fluorescence camera systems have become available in clinical practice. Intestinal perfusion at the anastomosis site can be assessed in real time with ICG fluorescence angiography (ICG-FA), which uses ICG and an NIR camera system that can visualize ICG fluorescence (Figure 1). This technique is compatible with minimally invasive surgery that inherently uses laparoscopy. To date, several studies have been conducted to evaluate whether intraoperative ICG-FA is an effective tool for assessing bowel perfusion and viability that leads to decreased rates of AL.

## 3. Subjective Assessment of Colon Perfusion Using ICG-Fluorescence Angiography

### 3.1. Non-Randomized Studies

In 2010, Kudszus et al. first reported the utility of ICG-FA in colorectal surgery. This was a single-center, retrospective, case-matched study of 402 patients with colorectal cancer. Herein, in the ICG-FA group, the planned transection line of the intestine was changed in 13.9% of cases due to poor fluorescence, and the incidence of grade C AL in this group was reduced by 4% compared to that in the control group (3.5% vs. 7.5%) [39].

In 2015, Jafari et al. reported the results of the PILLAR-II trial, a prospective, multicenter, clinical study of 139 patients who underwent left-sided colectomy and anterior resection [40]. In this study, the surgical plan was changed based on ICG-FA findings in 11 patients (8%), mostly in terms of changes in the planned transection line. Although this was a single-arm study, most of the patients included had benign diseases, cases of low anastomoses <5 cm from anal verge (AV) were excluded, and the diagnosis of AL was made only when clinically suspected and radiologically confirmed that the abscess was communicated with the anastomosis, the AL rate was low at 1.4%.

Since then, several clinical trials have examined the association between ICG-FA use and the occurrence of postoperative AL (Table 1).

In 2015, Kin et al. reported the results of a study of AL rates after colorectal resection in 173 pairs of case-matched patients (ICG-FA group vs. historical control), including those with benign and malignant disease [41]. In this study, AL was defined as an anastomotic defect or an abscess that was confirmed to communicate with anastomosis by physical or radiological examination. In the ICG-FA group, the planned transection line was changed in 4.6% of patients, but there was no significant difference in the AL incidence rate between the ICG-FA and control groups (7.5% vs. 6.4%, *p* = 0.67). Multivariate analysis showed that a distal anastomosis was significantly correlated with AL, while the use of ICG-FA was not.

In 2017, Kim et al. reported the results of a study on the usefulness of ICG-FA for postoperative AL reduction in 657 patients who underwent robot-assisted sphincter-saving surgery for rectal cancer [42]. In this study, the patients with anastomotic disruption that was confirmed by radiological examinations were diagnosed with AL. The unique feature of this study was that the authors assessed tissue perfusion by visually measuring the perfusion time and classifying the fluorescence intensity into five grades using a color standard. The AL rate was significantly lower in the ICG-FA group than in the control group (0.6% vs. 5.2%, *p* = 0.006). Furthermore, delayed perfusion (>60 s) and low perfusion intensity grade were significantly correlated with the occurrence of anastomotic stricture, an anastomosis-related complication (*p* < 0.001).

In 2018, Ris et al. conducted a multicenter, prospective, phase II study to assess whether ICG-FA could assist in the selection of intestinal transection levels and subsequent anastomotic vascular sufficiency [43]. This study included 504 patients who underwent colorectal resection for benign and malignant diseases and the definition of AL was Clavien-Dindo (CD) grade ≥3. The planned transection line was changed based on ICG-FA findings in 24 patients (4.7%), and in five of the 90 patients who underwent low anterior resection, a diverting stoma was not constructed based on the ICG-FA results. Therefore, the surgical plan was changed for 29 patients (5.8%) owing to the use of ICG-FA. The overall AL rate for colorectal resection in this study was 2.6%, which was significantly lower than the rate of 5.8% for 1,173 of the same procedures performed without ICG-FA at the participating centers (*p* = 0.009). The AL rate for right-sided colorectal resection was not significantly different between the ICG-FA and control groups (2.8% vs. 2.6%, *p* = 0.928), but this was significantly different for left-sided colorectal resection (2.6% vs. 6.8%, *p* = 0.005).

In 2019, Dinallo et al. conducted a retrospective study involving 554 colorectal resection cases with and without ICG-FA for benign and malignant diseases [44]. The patients with anastomotic disruption evidenced by post-operative imaging or endoscopy were diagnosed as AL. The study found no significant difference in AL rates between the ICG-FA and control groups (1.3% vs. 1.3%, *p* > 0.05); however, the transection line was changed in 5.6% of cases in the ICG group, which was significantly higher than that in the control group.

Foo et al. reported the results of a retrospective study of 506 patients who underwent rectal resection in 2020 [45]. In this study, most patients had rectal cancer, and the relationship between ICG-FA and postoperative AL, including all grades, was examined in a matched cohort using propensity scores. As a result of ICG-FA use, the transection line was changed in 20.9% of patients. The postoperative AL rate was significantly lower in the ICG-FA group than in the control group (3.6% vs. 7.9%, *p* = 0.035), and the percentages of grades A, B, and C AL were similar between the two groups (*p* = 0.760). Subgroup analysis showed significantly lower AL rates in the ICG-FA group than in the controls among those who underwent TME (4.7% vs. 11.6%, *p* = 0.043), but no significant difference was observed in patients who underwent non-TME (2.4% vs. 3.5%, *p* = 0.612).

In 2020, Hasegawa et al. reported the results of a retrospective study using propensity score matching in patients who underwent sphincter-sparing surgery for rectal malignancies [46]. In this study, an AL of CD grade ≥2 was defined as symptomatic AL. In the ICG-FA group, the transection line was changed in 17.0% of patients. The AL rate was significantly lower in the ICG-FA group than in the control group (2.8% vs. 13.6%, *p* = 0.001). Multivariate analysis showed that the use of ICG-FA also significantly reduced the risk of AL.

Watanabe et al. conducted a multi-enter retrospective study in 2020 [47]. This study included 422 patients with rectal cancer who underwent laparoscopic low anterior resection, and propensity score matching was used to balance the patient background between the ICG and control groups. Consequently, the intestinal transection line was changed to the oral side in 5.7% of patients. The AL rate in the ICG-FA group was significantly lower than that in the control group (CD grade ≥2: OR, 0.427; 95% CI, 0.197–0.926; *p* = 0.042; CD grade ≥3: OR, 0.280; 95% CI, 0.110–0.711; *p* = 0.007). In addition, the reoperation rates and duration of postoperative hospital stay were significantly lower in the ICG-FA group (OR, 0.192; 95% CI, 0.042–0.889; *p* = 0.036; and mean difference, 2.62 days; 95% CI 0.96–4.28; *p* = 0.002, respectively).

The results of other small-scale, non-randomized studies are shown in Table 1 [48,49,50,51,52,53,54,55,56,57,58,59,60].

Based on the results of the non-randomized studies described above, changes in the planned transection line due to poor fluorescence revealed by the use of ICG-FA are generally observed in approximately 10% (1.6–30.0%) of cases. Regarding the incidence of AL, the majority of reports concluded that the use of ICG-FA significantly reduced its incidence [42,43,45,46,47,49,52,53,56,57,58,59,60], whereas several studies failed to demonstrate a significant reduction in AL rates [41,44,48,50,51,54,55]. However, most of these studies were conducted at single centers, and there are many variations in the definition of AL, the target disease (malignant vs. benign), the type of surgery (right or left sided colon resection or rectal resection), and other background factors involved in AL occurrence. Factors related to the assessment of fluorescence, including the ICG dose and the NIR system used, were also inconsistent across studies. In addition, bias due to the retrospective nature of the studies needs to be considered when interpreting the results.

### 3.2. Randomized Controlled Trials

In 2020, De Nardi et al. reported the results of a randomized controlled trial (RCT) with three Italian participating institutions [61]. The study included 240 patients who underwent laparoscopic anterior rectal resection or left colectomy with colorectal anastomosis located 2–15 cm from the AV for either malignant or benign disease and all grades of AL were analyzed. In the ICG-FA group, ICG fluorescence of the colonic stump was assessed twice, before resection of the colon and after completion of the anastomosis. The results showed that the planned transection line was changed in 11% of cases in the ICG-FA group but they did not demonstrate a significant reduction in the AL rate compared to that of the control group (5% vs. 9%, *p* = 0.2).

Alekseev et al. reported in 2020 the results of the FLAG trial, a single-center RCT in Russia involving 377 patients who underwent sigmoid and rectal resection [62]. The results showed a significant reduction in the AL rate in the ICG-FA group compared to that in the control group (9.1 vs. 16.3%, *p* = 0.04), especially in 216 patients who underwent low anastomosis (AV, 4–8 cm) (14.4 vs. 25.7%, *p* = 0.04). However, these results were predominantly based on minor AL (grade A), which did not affect the patient’s postoperative course. Thus, no difference was observed in the reoperation rate and length of postoperative hospital stay. Therefore, the benefits of ICA-FA are limited.

In 2021, Jafari et al. reported the results of the PILLAR-III trial, a multicenter RCT involving 25 U.S. centers [63]. This study included 347 patients with rectal neoplasms who underwent low anterior resection with anastomoses <10 cm from the AV, and AL was defined as any evidence of endoluminal contents through the anastomosis as identified by imaging, drain output or at reoperation, or by endoscopic evidence of an anastomotic defect. The results showed no significant reduction in the AL rate in the ICG-FA group compared to the control group (9.0 vs. 9.6%, *p* = 0.37). However, notably, this clinical trial was statistically underpowered because it was terminated early due to slow recruitment; it did not achieve the pre-determined minimal sample size.

Therefore, the results of previous RCTs did not suggest that ICG-FA significantly reduced the AL rate. However, most recently, the authors of an RCT in Japan reported promising results at the 30th International Congress of Endoscopic Surgery in the European Union (EAES) in 2022 [64]. The study (EssentiAL study) was a multicenter RCT that included 41 Japanese centers and analyzed 839 patients who underwent sphincter-preserving surgery for rectal cancer. The results showed a significantly lower AL rate in the ICG-FA group than in the control group (7.6% vs. 11.8%, *p* = 0.041).

**Table 1 cancers-14-05623-t001:** Summary of the data from clinical studies investigating the reduction of anastomotic leakage with ICG-FA.

First Author	Year of Publication (Study Interval)	Country	Study Design	N (ICG/Control)	Age (ICG/Control)	Sex, Male (%) (ICG/Control)	Patient Type of Surgery	ICG Dose	NIR System	Transection Line Change (%)	AL (%) (ICG/Control)	*p* Value * for AL
Kudszus S [39]	2010 (1998–2008)	Germany	Retrospective, Single-center, Case-matched	402 (201/201)	67.8/69.0	42.2/42.2	Colorectal cancer (Rt, Lt, AR)	0.2–0.5 mg/kg	IC-View (Pulsion)	13.9	3.5/7.5	NA
Jafari MD [40]	2015 (2012–2013)	USA	Prospective, Multi-center (PILLAR-II)	139 (139/-)	58/-	53.2/-	Colorectal cancer Benign disease (Lap-Lt, AR)	3.75–7.5 mg/body	PINPOINT (Novadaq)	8	1.4/-	-
Kin C [41]	2015 (2005–2012)	USA	Retrospective, Single-center, Case-matched	346 (173/173)	58.1/58.2	54/54	Colorectal cancer Benign disease (Lt, AR)	7.5 mg/body	SPY Elite (Novadaq)	5	7.5/6.4	0.67
Boni L [48]	2017 (2012–2015)	Italy	Retrospective, Single-center	80 (42/38)	69/67	66.7/57.9	Colorectal cancer (Lap-LAR)	0.2 mg/kg	IMAGE1 (Karl Storz)	4.7	0/5	NS
Kim JC [42]	2017 (2010–2016)	Korea	Retrospective, Single-center	657 (310/347)	58/57	58.9/62.2	Rectal cancer (Rob-LAR, ISR)	10 mg/body	Da Vinci firefly	N.A.	0.6/5.2	<0.001
Brescia A [49]	2018 (2014–2017)	Italy	Prospective, Single-center	182 (75/107)	67.1/65.7	57.3/58.9	Colorectal cancer Benign disease (Lap-Rt, Lt, AR)	0.25 mg/kg	SPIES (Karl Storz)	6.6	0/5.6	0.03
Ris F [43]	2018 (2013–2016)	Switzerland Ireland UK	Prospective, Multi-center	1677 (504/1173)	64/NA	55.4/NA	Colorectal cancer Benign disease (Rt, AR)	7.5 mg/body	PINPOINT	5.8	2.6/5.8	0.009
Dinallo AM [44]	2019 (2010–2016)	USA	Retrospective, Single-center	554 (234/320)	61.5/62.5	45.2/43.1	Colorectal resection (Rt, Lt, LAR, total,)	5 mg/body	SPY Elite	5.6	1.3/1.3	>0.05
Shapera E [50]	2019 (2012–2018)	USA	Retrospective, Single-center	104 (74/30)	58/60	56.8/56.7	Colorectal cancer Benign disease (Rob-Lt, AR)	25 mg/body	Da Vinci firefly	5.4	0/3.3	0.289
Wada T [51]	2019 (2009–2016)	Japan	Retrospective, Single-center, PSM	68 (34/34)	67.5/66.5	58.8/70.6	Rectal cancer (Lap-LAR)	5 mg/body	PDE-neo (Hamamatsu Photonics)	27.1	8.8/14.7	0.71
De Nardi P [61]	2020 (2016–2017)	Italy	RCT, Multi-center	240 (118/122)	66.1/65.1	50.8/54.1	Colorectal cancer Benign disease (Lap-AR)	0.3 mg/kg	D-light P (Karl Storz)	11	5/9	0.2
Alekseev M [62]	2020 (2018–2019)	Russia	RCT (FLAG trial), Single-center	377 (187/190)	63/63	49.2/48.4	Colorectal cancer Benign disease (S and AR)	0.2 mg/kg	D-light P	19.3	9.1/16.3	0.04
Foo CC [45]	2020 (2013–2018)	China	Retrospective, Single-center, PSM	506 (253/253)	66.6/67.2	65.6/64.4	Rectal cancer Benign disease (Lap-AR)	5–7.5 mg/body	PINPOINT SPY Elite DaVinci firefly	20.9	3.6/7.9	0.035
Hasegawa H [46]	2020 (2007–2017)	Japan	Retrospective, Single-center	420 (141/279)	63/63	70.2/72.8	Malignant rectal tumor (Lap-LAR)	5 mg/body	IMAGE1 1588 AIM Hyper Eye	17.0	2.8/13.6	0.001
Impellizzeri HG [52]	2020 (2014–2019)	Italy	Retrospective, Single-center	196 (98/98)	66/71	55/58	Colorectal cancer Benign disease (Lt, AR, LAR)	12.5 mg/body	D-light P	8	0/6	0.029
Ishii M [53]	2020 (2014–2018)	Japan	Retrospective, Single-center, PSM	174 (87/87)	64/65	56.3/57.5	Rectal cancer (Lap/Rob-LAR, ISR)	5 mg/body	N.A.	3.1	3.4/11.5	0.044
Skrovina M [54]	2020 (2015–2017)	Czech Republic	Retrospective, Single-center	100 (50/50)	64/66	68/58	Rectal cancer (Lap/Rob-LAR, ISR)	0.2 mg/kg	SPIES Da Vinci Firefly	12	10/18	0.163
Tsang YP [55]	2020 (2018–2019)	China	Prospective, Single-center	131 (62/69)	69.8/67.7	62.9/68.1	Colorectal cancer Benign disease (Lt, AR, LAR)	10 mg/body	CLV-S200-IR (Olympus) Da Vinci Firefly	1.6	3.23/4.35	1.000
Watanabe J [47]	2020 (2014–2017)	Japan	Retrospective, Multi-center, PSM	422 (211/211)	66/66	60.7/62.1	Rectal cancer (Lap-LAR)	0.25 mg/kg	1588 AIM (Stryker) D-light P	5.7	4.7/10.4	0.042
Wojcik M [56]	2020 (2017–2018)	France	Prospective, Case-matched	84 (42/42)	67/69	69/69	Colorectal cancer (Lap-AR, LAR)	0.1 mg/kg	PINPOINT	10.9	2.4/16.7	0.026
Jafari MD [63]	2021 (2015–2017)	USA	RCT (PILLAR-III), Multi-center	347 (178/167)	57.2/57.0	61.2/58.6	Rectal cancer (AR)	5–10 mg/body	PINPOINT SPY Elite	N.A.	9.0/9.6	0.37
T Yanagita [57]	2021 (2011–2018)	Japan	Retrospective, Single-center, PSM	186 (93/93)	NA/NA	NA/NA	Colorectal cancer (Lt, AR)	0.1 mg/kg	Hyper Eye (Mizuho) IMAGE1	9.1	3.2/10.8	0.046
Otero-Pineiro AM [58]	2021 (2011–2018)	Spain	Retrospective, Single-center	284 (80/204)	68.0/66.6	63.7/60.3	Rectal cancer (taTME-AR)	N.A.	PINPOINT	28.7	2.5/11.3	0.020
Hasegawa H [59]	2022 (2010–2016)	Japan	Retrospective, Single-center, PSM	169 (66/103)	NA/NA	75.8/78.6	Rectal cancer (Lap-ISR)	5 mg/body	IMAGE1 SPY Hyper Eye	30.0	0/14.6	0.001
Neddermeyer M [60]	2022 (2017–2020)	Germany	Retrospective, Single-center	132 (70/62)	66.5/59.5	68.6/62.9	Colorectal cancer (S, AR)	25 mg/body	PINPOINT	12.9	1.4/14.5	0.006

ICG, indocyanine green; NIR, near-infrared; RCT, randomized controlled trial; PSM, propensity score matching; Lap, laparoscopic; rob, robotic; Rt, right colectomy; Lt, left colectomy; S, sigmoidectomy; total, total colectomy; AR, anterior resection; LAR, low anterior resection; ISR, intersphincteric resection; NA, not available; NS, not significant. * Statistical analyses were performed using Fisher’s exact test or Chi-square test as appropriate.

### 3.3. Discussion of Clinical Studies with Subjective Assessment of ICG-Fluorescence Angiography

There were inconsistencies between the results of the several non-randomized observational studies that have shown that ICG-FA reduces AL rates and those of the RCTs that have not demonstrated this finding. It was difficult to draw clear conclusions, as the former may involve inherent biases, mainly due to retrospective case-matched single-center studies, and the latter may involve a lack of statistical power. However, the results of a recent RCT are encouraging. The EssentiAL study was well designed and achieved enrollment of a predetermined number of patients in a relatively short period of time; thus, its results, which show that ICG-FA is useful in reducing AL, are credible [64]. In addition, several other large-scale and well-designed RCTs are currently underway, and their results are awaited [65,66]. If the results of these RCTs replicate those of the EssentiAL trial, definitive conclusions will be drawn regarding whether ICG-FA reduces AL rates in colorectal surgery.

## 4. Quantitative Assessment of Colon Perfusion Using ICG-Fluorescence Angiography

### 4.1. Limitations of Subjective Assessment

Compared to traditional methods of evaluating blood perfusion of the intestinal stump, such as observation of intestinal wall color, visible peristalsis, bleeding from the marginal artery, and tactile detection of mesenteric artery pulsation, ICG-FA clearly highlights perfused tissue with fluorescence, thereby improving the reproducibility and reliability of blood perfusion assessment. This allows the surgeon to select the transection line of the intestine intuitively. However, the major limitation of this method is that the evaluation of fluorescence intensity is subjective, based on visual assessment by the surgeon.

If blood perfusion is maintained in the intestinal stump, fluorescence is usually visible within 10–60 s after ICG administration from a peripheral vein, and a demarcation is formed in the area where blood flow is lost. At this point, if the demarcation at the planned transection line is clear, the transection line can be easily relocated in an area where perfusion is maintained. However, the lack of uniform evaluation criteria for blood perfusion when the fluorescence is faint or the demarcation line is obscure is a major issue. Reportedly, ICG fluorescence assessment is prone to be interpreted variably by different observers and is also influenced by the surgeon’s experience [67]. Therefore, the development of quantitative fluorescence parameters is necessary to reduce the subjective element in surgical decision-making.

### 4.2. Methods for Quantitative Assessment of ICG-Fluorescence Angiography

Quantitative assessment of fluorescence in ICG-FA is commonly performed by calculating the various quantitative parameters by analyzing changes in fluorescence intensity using a computer algorithm and converting them into a fluorescence-time curve (Figure 2). These quantitative parameters can be classified into two categories. One is the *intensity* parameter, which is represented by the maximum fluorescence intensity (F_max_: difference in fluorescence between the maximum and baseline). The other is the *inflow* parameter, which includes T_0_ (time from ICG infusion to first fluorescence signal), T_max_ (time from ICG infusion to maximum fluorescence), T_1/2max_ (time from first to half maximum fluorescence signal), time-to-peak (ttp: time from first to maximum fluorescence signal, calculated by T_max_ − T_0_), slope (F_max_/ttp), and the time ratio (TR: T_1/2max_/ttp).

### 4.3. Clinical Studies with Quantitative Assessment of ICG-Fluorescence Angiography

To date, six studies have reported whether the aforementioned quantitative parameters correlate with the occurrence of postoperative AL in colorectal surgery [68,69,70,71,72,73]. A list of these clinical studies is presented in Table 2.

Wada et al. reported the results of a retrospective analysis of 112 cases of laparoscopic surgery for left-sided colorectal cancer [68]. Herein, when the cutoff value of F_max_ was 52.0 AU, the sensitivity and specificity for the prediction of AL were 100% and 92.5%, respectively (positive predictive value (PPV), 38.5%; negative predictive value (NPV), 100%). Similarly, when the cut-off value of the slope was set at 2.0 AU/s, the sensitivity and specificity for AL prediction were 100% and 75.7%, respectively (PPV: 16.1%, NPV: 100%). Receiver operating characteristic (ROC) analysis showed that F_max_ and slope were predictive of AL, but ttp and T_1/2max_ were not.

Son et al. investigated whether each quantitative parameter was correlated with postoperative anastomotic complications in 86 patients with colorectal cancer who underwent laparoscopic anterior resection [69]. The results showed that F_max_, an *intensity* parameter, was not significantly different between the postoperative anastomotic complication group and the control group (34.9 vs. 58.0, *p* = 0.074), but the *inflow* parameters were significantly different between the two groups (ttp: 64.0 vs. 30.3, *p* < 0.001; T_1/2max_: 40.4 vs. 11.7, *p* < 0.001; Slope: 0.7 vs. 2.5, *p* < 0.001; TR: 0.6 vs. 0.4, *p* < 0.001). In the ROC analysis, T_1/2max_ and TR showed significant predictive values for anastomotic complications.

Hayami et al. reported the results of a prospective study of 22 patients with colorectal cancer [70]. In this study, only the *inflow* parameter T_0_ was significantly longer in the AL group than in the control group (64.3 vs. 18.2, *p* = 0.0022), but other parameters showed no significant between-group differences (F_max_: 79.9 vs. 0.4, *p* = 0.42; ttp: 26.4 vs. 18.6, *p* = 0.09; T_1/2max_: 13.3 vs. 7.8, *p* = 0.12; Slope: 3.4 vs. 5.5, *p* = 0.27).

Iwamoto et al. reported the results of a prospective study of 25 patients with rectal cancer at the same institution. Similar to the results of the study by Hayami et al., T_0_ was significantly correlated with the occurrence of AL (37.5 vs. 11.0, *p* = 0.03), whereas ttp was not (11.5 vs. 12.0, *p* = 0.85) [71].

Amagai et al. reported the results of a prospective study of 69 patients who underwent left-sided colorectal resections [72]. They observed fluorescence from the luminal side with a transanally inserted NIR scope and evaluated blood flow at the anastomotic site by setting the regions of interest (ROIs) at the highest and lowest fluorescence areas of the proximal and distal intestine, respectively. The results showed that F_max_ was not significantly correlated with AL at any of the four measurement points. In contrast, T_max_ and ttp in the lowest fluorescent region were significantly correlated with AL (74.3 vs. 45.4, *p* = 0.015 and 52.8 vs. 30.0, *p* = 0.040, respectively).

Gomez-Rosado et al. reported that a lower F_max_ and slope were significantly correlated with AL in a retrospective study of 70 colorectal cancer surgery cases (151 vs. 169, *p* = 0.03 and 10.8 vs. 17.4, *p* = 0.0.3, respectively). In contrast, T_max_ and T_0_ were not significantly correlated with AL occurrence (47.6 vs. 35.5, *p* = 0.10 and 19.9 vs. 19.8, *p* = 0.98, respectively) [73].

### 4.4. Discussion of Clinical Studies with Quantitative Assessment of ICG-Fluorescence Angiography

Based on these results, quantitative assessment of ICG-FA seems to be a promising method for identifying patients at a high risk for AL. In particular, any of the *inflow* parameters (ttp, T_max_, T_1/2max_, T_0_, slope, TR) showed significant differences between AL and non-AL patients in all studies listed in Table 2. However, the accuracy of each parameter as a predictor of AL is inconsistent across studies. On the other hand, the results for F_max_, an *intensity* parameter, were not consistent, and only two of the five studies examined found significant differences between AL and non-AL patients, while the other three did not.

In the quantitative evaluation of ICG-FA, the *intensity* parameter has been reported to show a strong positive correlation with tissue oxygen saturation, which is important in the healing process of anastomotic tissue [74]; at first glance, it seems to be a useful quantitative parameter. However, this parameter may not be a strong predictor of clinical AL because of the following inherent problems. First, the fluorescence intensity is highly influenced by external factors, such as the camera angle and distance to the object, background light, equipment used, and ICG concentration in the blood. Fluorescence intensity is inversely correlated with the distance between the camera and the tissue and is very sensitive to slight changes in this distance; thus, the slightest jolt of the handheld camera can lead to noise and interference with the quantitative analysis [75,76]. In addition, previous studies used different protocols for ICG dosage (Table 1 and Table 2). Fluorescence intensity has been found to be affected by blood ICG concentration [76]. Although it is impossible to obtain identical blood concentrations in each patient because of the differences in drug metabolism, cardiac output, vascular status, and volume of distribution, protocols with fixed ICG doses may result in large variations in fluorescence intensity due to differences in patients’ body mass index. Therefore, it is recommended to use a dose calculated based on body weight [77]. In addition, each previous study used a different NIR system, which is known to affect fluorescence intensity measurements [76]. As fluorescence emission sources, lasers have been reported to provide higher resolution and fluorescence intensity than xenon lamps; therefore, this difference should also be noted. Another problem with the *intensity* parameter is that it can cause overestimation of tissue perfusion. Fluorescence intensity in the perfusion area usually peaks within 60 s after ICG administration and then reaches a plateau (Figure 2); however, over time, capillary diffusion may cause fluorescence to be observed even in ischemic areas [78].

In contrast, the *inflow* parameter appears to be less sensitive to such external factors because it depends primarily on the time axis rather than the absolute value of the fluorescence intensity. In fact, the values of each *inflow* parameter are highly reproducible across different studies, and the variation in each value in the non-AL group appears to be relatively small (Table 2). Thus, the *inflow* parameter represents dynamic changes in tissue perfusion more objectively than the *intensity* parameter, which may be one of the reasons for its greater accuracy as a predictor of postoperative AL.

It should be noted, however, that previous studies on the quantitative evaluation of ICG-FA do not involve standardized protocols and their results are from small-scale, non-randomized, single-center studies, with a low level of evidence. In addition, most current quantitative assessments of ICG-FA are based on postoperative evaluations of video recordings of surgical procedures. Therefore, one of the limitations is that the analysis results cannot be used for real-time decision-making by surgeons. Currently, among the commercially available NIR systems, the SPY Elite system (Novadaq Technologies Inc., Bonita Springs, FL) can quantify fluorescence intensity in real time with built-in software, but only the *intensity* parameter can be evaluated [79].

Therefore, large-scale clinical studies with appropriately standardized protocols are needed to establish the usefulness of quantitative assessment of ICG-FA. In addition, it is desirable to introduce technology that enables real-time quantification of fluorescence, including the *inflow* parameter.

## 5. Future Perspectives

While ICG-FA is feasible as an evaluation method for tissue perfusion with minimal side effects, the currently used subjective assessment based on surgical visualization has limitations, especially when the fluorescence is obscure. To complement this, novel evaluation methods using various quantitative parameters have been proposed; however, as each patient may exhibit an individual fluorescence-time curve shape, their interpretation remains subjective, requires training, and can be a source of error, especially for inexperienced surgeons.

To overcome these limitations, artificial intelligence (AI) is expected to provide a more objective evaluation method. AI is currently being applied in medical practice and its implementation in blood flow assessment using ICG-FA is under consideration [80,81]. Park et al. showed that AI-based real-time microcirculation analysis is more accurate and consistent than conventional parameter-based analysis [82]. Once AI, which can be automatically and more objectively assessed, is fully implemented in this field, surgeons will be able to reliably predict the risk of anastomotic complications due to hypoperfusion.

ICG-FA has its own drawbacks, including the associated need to administer external fluorophores and limited number of repeat assessments. It also cannot be used in patients with an iodine allergy. Novel technologies may overcome these problems in the future. For instance, laser speckle contrast imaging (LSCI), which specifically displays real-time blood flow and tissue perfusion by detecting the movement of objects such as red blood cells using coherent laser light, has advantages over ICG-FA: it facilitates repeat assessments, precludes the administration of additional drugs, and involves no latency from dye injection to perfusion [78]. In a study by Liu et al., LSCI showed high concordance with ICG-FA in the assessment of intestinal perfusion in human participants, and was associated with high surgical convenience [83]. Furthermore, this technique has the advantage of eliminating false positives caused by capillary diffusion, which spreads fluorescence to the ischemic region over time, as observed in ICG-FA.

Modern surgeries have evolved with advances in technology. Advances in surgical devices, including high-performance imaging systems, are expected to improve the safety of colorectal surgery, which in turn will benefit patients, physicians, and society as a whole.

## 6. Conclusions

ICG-FA is an innovative technique that allows the visualization of tissue perfusion. Based on this information, surgeons will be able to clearly identify intestinal segments with good blood flow for safer colorectal anastomosis; therefore, this technique is now widely applied in clinical practice. However, the results of clinical trials investigating whether ICG-FA reduces the risk of AL in colorectal surgery have been inconsistent. Several large-scale RCTs are currently underway, and their results will determine whether ICG-FA is, indeed, useful. In addition, several methods have been proposed to objectively evaluate fluorescence using various quantitative parameters. Optimal standardization of protocols is expected to improve the accuracy and reliability of ICG-FA.

Once the usefulness of this new imaging technology is established, it will make a significant contribution to this field by improving the safety and outcomes of colorectal surgery.

## Figures and Tables

**Figure 1 cancers-14-05623-f001:**
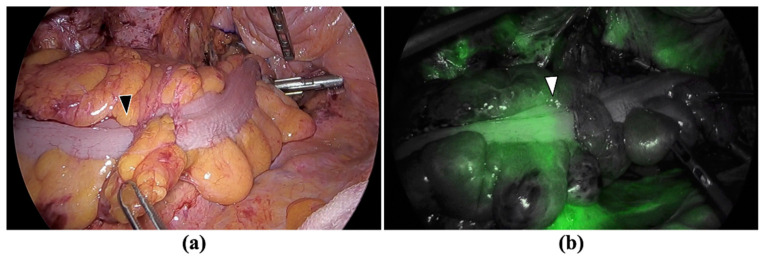
Perfusion assessment of colonic stump with ICG-FA: (**a**) Observation under white light just before anastomosis. There appears to be no difference in the coloration of the intestine between the oral and anal sides of the black arrowhead; (**b**) Observation with the NIR camera after ICG injection revealed that there was no vascular perfusion on the anal side from the white arrowhead.

**Figure 2 cancers-14-05623-f002:**
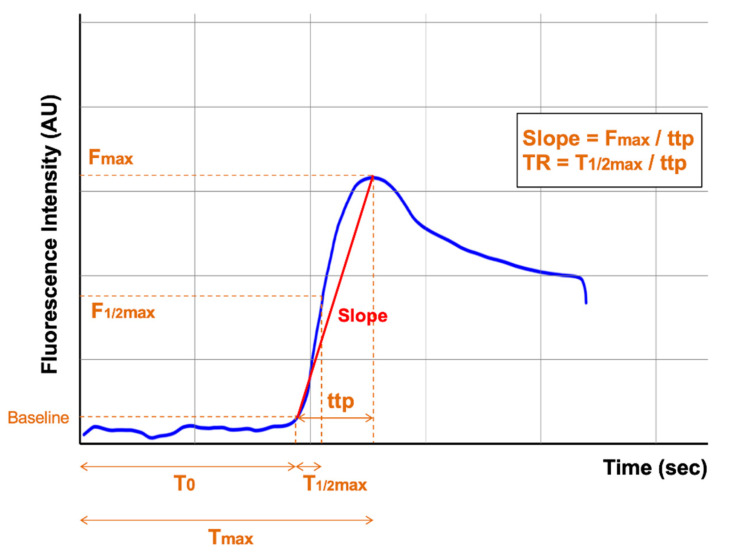
Fluorescence-time curve and quantitative parameters of ICF-FA. AU: arbitrary unit, F_max_: maximum fluorescence intensity, F_1/2max_: half maximum fluorescence intensity, T_0_: time from ICG injection to first fluorescence signal, T_max_: time from ICG injection to maximum fluorescence signal, T_1/2max_: time from first to half maximum fluorescence signal, ttp (time-to-peak): T_max_ − T_0_, Slope: F_max_/ttp, TR (the time ratio): T_1/2max_/ttp.

**Table 2 cancers-14-05623-t002:** Summary of clinical studies with quantitative assessment of ICG-FA.

First Author	Year	ICG Dose	Imaging System (Analysis Software)	Intensity Parameter	AL	Non-AL-	p Value	Inflow Parameter	AL	Non-AL-	*p* Value
Gomez-Rosado JC [73]	2021	7.5 mg /body	Elevision^TM^ IR Platform (Not shown)	F_max_ (AU)	151 (±13.1)	169 (±24.0)	0.03	T_max_ (s) Slope (AU/s) T_0_ (s)	47.6 (±20.7) 10.8 (±6.2) 19.9 (±14.3)	35.5 (±16.6) 17.4 (±7.4) 19.8 (±11.8)	0.10 0.03 0.98
Amagai H [72]	2020	0.2 mg /kg	Olympus (Image J)	F_max_ (AU)	Not shown	Not shown	0.380	ttp (s) T_max_	52.8 (±30.3) 74.3 (±42.3)	30.0 (±15.0) 45.4 (±21.4)	0.040 0.015
Iwamoto H [71]	2020	7.5 mg /body	PINPOINT (ROIs)	-	-	-	-	ttp (s) T_0_ (s)	11.5 (±7.3) 37.5 (±17.1)	12.0 (±9.3) 11.0 (±13.1)	0.85 0.03
Hayami S [70]	2019	5 mg /body	D-light P (ROIs)	F_max_ (AU)	79.9 (±28.5)	87.6 (±33.2)	0.42	ttp (s) T_1/2max_ (s) Slope (AU/s) T_0_ (s)	26.4 (±8.4) 13.3 (±4.9) 3.4 (±2.0) 64.3 (±27.6)	18.6 (±6.2) 7.8 (±2.9) 5.5 (±2.8) 18.2 (±6.6)	0.09 0.12 0.27 0.0022
Son GM [69]	2019	0.25 mg /kg	IMAGE1 S^TM^ (Tracker 4.97)	F_max_ (AU)	34.9 (±7.4)	58.0 (±3.4)	0.074	ttp (s) T_1/2max_ (s) Slope (AU/s) TR	64.0 (±11.7) 40.37 (±7.8) 0.7 (±0.2) 0.6 (±0.0)	30.3 (±2.3) 11.7 (±0.8) 2.5 (±0.2) 0.4 (±0.0)	<0.001 <0.001 <0.001 <0.001
Wada T [68]	2017	5 mg /body	PDE-neo (ROIs)	F_max_ (AU)	38.1 (±11.4)	91.4 (±31.9)	NA	ttp (s) T_1/2max_ (s) Slope (AU/s)	52.1 (±28.5) 26.1 (±18.9) 0.98 (±0.7)	32.8 (±15.9) 12.5 (±7.6) 3.6 (±2.2)	NA NA NA

ICG, indocyanine green; AL, anastomotic leak; AU, arbitrary unit; NA, not available

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
