# Peer review of "A Narrative Review of the Usefulness of Indocyanine Green Fluorescence Angiography for Perfusion Assessment in Colorectal Surgery"

_cancers, 2022, doi:10.3390/cancers14225623_

Round 1

Reviewer 1 Report

Excellent review of this subject, very good review of the literature, of current practice, and of limitations of this technology.

page 6, line 206:  language issue I think.  Do you mean "studies" not just "study"?  Otherwise I am not sure to which one study you are referring.

page 6, line 209: language issue I think.  "predominant report" is not correct use here.

p12, line 457 and 461:  I believe "ICF-FA" and "ICFA-FA" should both be "ICG-FA"

Reviewer 2 Report

Dear Authors, 

I had the chance to review your manuscript titled "A Narrative Review of the Usefulness of Indocyanine Green Fluorescence Angiography for Perfusion Assessment in Colorectal Surgery".

As a narrative review, the paper is well structured and adequately developed.

In my opinion, the introduction, as regards principles of use of Indocyanine Green and anastomotic leakage etiopathogenesis, could be more concise, as this is not an original study but a review on a known topic extensively treated in literature. 

An interesting element of novelty is represented by the analysis of literature on quantitative assessment of perfusion of ICG fluorescent angiography (section 4). This subtopic has been well explained and results clearly reported. 

On the other hand I don't think it is appropriate to summarize in this narrative review the results of existing metanalysis as it has been done in section 3.4 and table 2. Metanalysis on this topic are often large and complex papers with different endpoints and different selection criteria, therefore it is not proper to report these results in a single paragraph. 

Reviewer 3 Report

Overall, the narrative review gives a good overview of current literature, corresponding NIRF systems used and important parameters. There is clearly described why ICG can be useful in assessing anastomotic perfusion as wel as the current limitations. To conclude, there is an accurate assessment of what's important for future perspectives to optimize and integrate ICG use in daily practice. 

In section 2.1 risk factors are described; include some extra known factors like age, alcohol abuse and intraoperative septic conditions. To finetune table 1: add study interval, countries, and age/gender as these are known risk factors. Additionally, please report the statistical tests on which the p values are based in the food note. Some studies report both oncological and benign cases; be aware of the fact that benign diseases (like IBD) have multiple extra risk factors on anastomotic healing. It can be good to split benign and malign cases in the tables to see if this has an influence on AL rates. 

The definitions of AL (and possible grading systems) used in the included studies are missing. As there is a huge variety between AL definitions, grading systems and reporting methods, it is from added value to integrate this information in your narrative review. E.g. in section 3.2. Alekseev et al. report their results based on minor AL (grade A; based on what?), which already indicates there is need for consistency in defining and reporting AL as this influences the results. 

In section 4.2 the methods for quantitative assessment of ICG-fluorescence angiography are described well. Table 2 gives an extra overview of studies reporting these parameters, but are none of the previous articles (included in table 1) mentioning data like time to peak, etc.? Otherwise, good to add as well. 

Optional, add contrast to other dyes in the introduction: why is ICG the one to use in bowel perfusion assessment and the one we need to optimize (e.g. compared to methylene blue or other currently investigated dyes).  
